# The Influence of School Atmosphere on Chinese Teachers’ Job Satisfaction: The Chain Mediating Effect of Psychological Capital and Professional Identity

**DOI:** 10.3390/bs13010001

**Published:** 2022-12-20

**Authors:** Xinqiang Han, Qian Xu, Junhu Xiao

**Affiliations:** School of Education Science, Shanxi Normal University, Taiyuan 030000, China

**Keywords:** school atmosphere, psychological capital, professional identity, job satisfaction

## Abstract

Until today, the impact of organizational atmosphere on job satisfaction has still attracted the attention of researchers in the field of education to help decision-makers and school leaders improve the teachers’ commitment, professional identity, and job satisfaction through the construction of the school environment. This study explored the impact of school atmosphere on the teachers’ job satisfaction and examined the chain mediating role of psychological capital and professional identity. The school atmosphere, psychological capital, professional identity, and job satisfaction scales were used to investigate 648 primary and secondary school teachers in China. The bootstrap method was used to test the mediating effect. The results showed that school atmosphere had a positive predictive effect on the teachers’ job satisfaction; psychological capital does not play a mediating role between school atmosphere and job satisfaction; professional identity plays a mediating role between school atmosphere and job satisfaction; psychological capital and professional identity play a chain mediating role between school atmosphere and job satisfaction. Therefore, this study proposes that schools adopt more effective school management strategies to build a positive school atmosphere to improve the teachers’ psychological capital and professional identity to solve the practical problem of low job satisfaction among primary and secondary school teachers.

## 1. Introduction

After the outbreak of novel coronavirus pneumonia in 2019, Chinese schools suddenly switched from the traditional school learning mode to the online education mode. During online teaching, as teachers needed to learn new digital technologies to help online teaching run more smoothly, the teachers’ workload increased compared with that before, and the quality of the teachers’ work became increasingly demanding [1]. When the pressure from the high workload and level of job requirements acts on teachers, teachers are likely to have a sense of frustration due to their limited resources, which may even directly lead to a significant decline in their job satisfaction. However, when the teachers’ job satisfaction is seriously reduced, it is more likely to cause teachers to implement teaching activities inefficiently and affect the regular teaching order [2].

The school atmosphere is a common psychological perception. This common psychological perception can promote mutual trust and help managers and teachers, teachers and teachers, to achieve the training goals set by the school for students and improve the teachers’ job satisfaction and sense of teaching mission. However, in the form of online teaching, the promotion of school atmosphere for teachers is very limited.

As far as practical work is concerned, due to online teaching, teachers must constantly fight against technical failures during video conferences, which makes them feel that their work tasks have become more tedious, and their work tension is more and more apparent. However, teachers cannot properly solve these problems [3]. Because of the epidemic, they cannot promptly seek support and comfort from schools and colleagues as they did before the epidemic. Teachers cannot ease tension for such a long time, likely leading to the continuous loss of psychological capital. Over time, teachers are prone to insomnia, tension, irritability, fatigue, and other negative emotions, leading to a decline in their professional identity and, ultimately, their job satisfaction. In the long run, this will affect the regular operation of school teaching, especially in the current unique situation. Therefore, it is necessary to explore the role of psychological capital and professional identity in the impact of school atmosphere on the teachers’ job satisfaction.

In the research on primary and secondary school teachers’ professional problems, only some studies have simultaneously examined the impact of environmental factors and personal factors on the teachers’ job satisfaction. This study further explored the mediating role of psychological capital and the professional identity of primary and secondary school teachers in the school atmosphere, affecting the teachers’ job satisfaction.

To sum up, a good school atmosphere can strengthen the trust between managers and teachers and reach a consensus so that teachers can persevere when encountering obstacles and come up with solutions at will, thus improving their psychological capital level [4,5]. Furthermore, teachers with high-level psychological capital can invest more energy and time in their work to learn and develop their potential, experience the fun of teaching, effectively improve their professional identity [6], and ultimately make them more satisfied with their work [7,8]. Therefore, it is not enough to just discuss job satisfaction to improve the current job satisfaction of primary and secondary school teachers. It also needs to consider the impact of the school atmosphere, the teachers’ psychological capital, and professional identity. In order to verify the feasibility of this measure, it is necessary to study the job satisfaction of primary and secondary school teachers and determine the impact of school atmosphere, psychological capital, and professional identity on job satisfaction. On this basis, it is possible to propose practical measures to ensure the effectiveness of the teachers’ work and provide a basis for further improving and managing the job satisfaction of primary and secondary school teachers.

## 2. Literature Review and Theoretical Background

### 2.1. Job Satisfaction

Teachers’ job satisfaction has many important and far-reaching implications [9]. It refers to the overall feelings of individual teachers regarding their work conditions [9]. It can significantly predict teacher retention and performance [10]. The job satisfaction of primary and secondary school teachers is a crucial factor affecting their work attitude and behavior [11]. At present, scholars around the world have accumulated a large number of fruitful results in research on the job satisfaction of primary and secondary school teachers. This study focused on the environmental factors and personal characteristics that affect the job satisfaction of primary and secondary school teachers [12]. For example, Zeeshan believes that the main factors that affect teacher satisfaction are school management, work environment, salary, colleague relations, work itself, safety, and other factors [13]. Lam investigated the factors that affected the teachers’ job satisfaction and proposed that school size, working hours, leadership, work achievements, and other factors will affect their job satisfaction [14]. Shaukat discussed the impact of gender, age, background, teaching experience, and professional qualifications of special education teachers in Pakistan on their job satisfaction. The results showed that the teachers’ gender, age, educational background, teaching experience, and other characteristics significantly impact job satisfaction [15]. Teachers’ job satisfaction affects their work status, teaching effectiveness, and quality of talent training [16].

Currently, the job satisfaction of primary and secondary school teachers in China is generally low [17], which seriously hinders the enthusiasm of teachers, the smooth development of teaching activities, and restricts the practical improvement of education quality [18]. Therefore, it is crucial for primary and secondary school teachers to improve job satisfaction, whether from the perspective of strengthening the construction of teachers, improving the working status of teachers, or the growth and development of students.

### 2.2. School Atmosphere and Job Satisfaction

The school atmosphere is the teacher’s subjective experience of the school’s norms, objectives, values, interpersonal relationships, teaching and learning practices, and organizational structure. It is an environmental feature that has a relatively lasting and stable impact on the teachers’ behavior [19]. School atmosphere can be divided into organization management, team cooperation, teaching efficiency, and resource use [20]. Research shows that the individual’s perceptions of the organizational environment significantly affect their job satisfaction [21,22], and further affect their job engagement and performance [23]. Social information processing theory believes that the social environment in which individuals live can affect their attitudes and beliefs and thus affect their perception of working conditions and states [24]. In addition to individual personality, the job satisfaction of organizational members is also affected by their perceived organizational culture. A positive organizational culture helps improve the organizational members’ job satisfaction [25]. Therefore, as teachers perceive organizational culture in their work situations, the school atmosphere can affect the teachers’ subjective feelings and job satisfaction. Therefore, this study hypothesizes that:

**Hypothesis** **1:**
*School atmosphere positively predicts teachers’ job satisfaction.*


### 2.3. The Mediating Role of Psychological Capital

Psychological capital is an individual’s positive psychological quality and a psychological resource to promote the formation of positive self-cognition, mainly including self-confidence, optimism, hope, and resilience [26]. Research has shown that psychological capital has a significant positive impact on job satisfaction and happiness [27,28,29,30]. Individuals with a high level of psychological capital are more optimistic and confident about their work. They can effectively cope with the challenges and difficulties at work, so they have a high degree of job satisfaction [29]. In addition, the school atmosphere is an essential factor affecting psychological capital. Organizational atmosphere can provide favorable conditions for improving individual psychological capital [26]. The organizational atmosphere focusing on employee training and development can effectively enhance their employees’ self-confidence and psychological resilience [31]. A flexible organizational atmosphere can enhance an individual’s confidence in their abilities and psychological resilience [32]. Specifically, a good school atmosphere can help teachers build experience advantages from successful team experiences [33]. It also strengthens team cohesion from a working atmosphere of cooperation and mutual trust [33], improves the teachers’ ability to work under pressure, and maintains an optimistic attitude toward work [29]. Therefore, a good school atmosphere positively affects psychological capital, and psychological capital also helps improve job satisfaction. Therefore, this study hypothesizes that:

**Hypothesis** **2a:**
*School atmosphere has a positive predictive effect on the teachers’ psychological capital;*


**Hypothesis** **2b:**
*Psychological capital has a positive predictive effect on the teachers’ job satisfaction.*


### 2.4. The Mediating Role of Professional Identity

Professional identity emphasizes the internalization of individual professional values and psychological acceptance and recognition of their professional identity [34]. Individuals with strong professional identity tend to be proud of their careers [35]. They have higher self-determination motivation, more enthusiasm and energy for work [36], lower job burnout [37], and higher job satisfaction and happiness [38,39,40]. In addition, the school atmosphere is an important factor affecting professional identity. An intense school atmosphere helps improve individual self-worth and promote professional identity development [41]. Specifically, reasonable teacher evaluation and reward mechanisms, humanistic care for teachers, tolerance, and openness of the school atmosphere, and the teacher culture are important factors affecting the professional identity of teachers [42]. Meanwhile, the norms and practices in the school atmosphere imperceptibly affect the teachers’ views on teaching and teacher identity [41]. Therefore, a good school atmosphere positively affects professional identity, and professional identity also helps improve job satisfaction. Thus, this study hypothesizes that:

**Hypothesis** **3a:**
*School atmosphere has a positive predictive effect on the teachers’ professional identity;*


**Hypothesis** **3b:**
*Professional identity has a positive predictive effect on the teachers’ job satisfaction.*


### 2.5. The Chain Mediating Role of Psychological Capital and Professional Identity

Psychological capital can effectively enhance professional identity and can affect occupational happiness through the mediation role of professional identity [43,44,45]. As a positive psychological resource for coping with the source of stress, psychological capital can stimulate more work engagement from teachers [46]. Psychological capital can enable teachers to adapt to the workplace environment and career development with a more proactive attitude and stress-coping style, and experience more professional joy [45]. Therefore, psychological capital helps teachers shape their positive professional values and behavior tendencies and effectively enhances their professional identity [45]. Therefore, a good school atmosphere can effectively improve job satisfaction through the mediating role of psychological capital and professional identity. Therefore, this study hypothesizes that:

**Hypothesis** **4:**
*Psychological capital and professional identity play a chain intermediary role between school atmosphere and job satisfaction.*


## 3. The Present Study

The theoretical model is shown in Figure 1. Previous studies have examined the impact of positive qualities (e.g., psychological capital and professional identity) on the teachers’ job satisfaction. However, few studies have focused on whether the school atmosphere can improve the teachers’ job satisfaction by enhancing their psychological capital and professional identity. Therefore, this study attempts to answer the question of “how environmental factors (such as school atmosphere) affect the teachers’ job satisfaction” by examining the mediating chain role of psychological capital and professional identity in school atmosphere and the teachers’ job satisfaction. This will help provide empirical evidence for schools to improve the job satisfaction of teachers.

## 4. Materials and Methods

### 4.1. Participants

A total of 785 questionnaires were distributed to primary and secondary school teachers in China by a convenient sampling method. After deleting invalid data with too short a response time, 648 valid data were recovered, with an effective rate of 82.55%.

### 4.2. Ethical Consideration

This study was conducted following the Declaration of Helsinki and Measures for Ethical Review of Biomedical Research Involving Humans, Ministry of Health, China. The protocol was approved by the Ethics Committee of Shanxi Normal University

### 4.3. Measures

School Atmosphere. This study measured the school atmosphere with fourteen items compiled by Ding [20]. The scale has four subscales: organization management, team cooperation, teaching efficiency, and resource utilization. Sample items of school atmosphere such as ‘The school will praise and encourage active and excellent teachers’ and ‘The school has many initiatives to encourage teaching reform and innovation’. The responses to statements were rated on a 5-point Likert scale (from 1 = strongly disagree to 5 = strongly agree). Cronbach’s αs for the total scale and subscales were 0.94, 0.81, 0.87, 0.91, and 0.87, respectively. CRs for the total scale and subscales were 0.95, 0.82, 0.87, 0.92, and 0.89, respectively. AVEs for the subscales were 0.52, 0.53, 0.62, 0.79, and 0.74, respectively. Using CFA to test the construct validity, χ^2^/df = 4.89, TLI = 0.95, CFI = 0.96, RMSEA = 0.07, RMR = 0.02, GFI = 0.93, PNFI = 0.70.

Psychological Capital. This study measured psychological capital with twenty-four items compiled by Luthans [47]. The scale has four subscales: confidence, optimism, resiliency, and hope. Sample items of psychological capital such as ‘I am optimistic about what will happen to my work in the future’ and ‘I usually take the pressure at work calmly’. The responses to statements were rated on a 5-point Likert scale (from 1 = strongly disagree to 5 = strongly agree). Cronbach’s αs for the total scale and subscales were 0.98, 0.94, 0.93, 0.94, and 0.94, respectively. CRs for the total scale and subscales were 0.98, 0.94, 0.94, 0.93, and 0.95, respectively. AVEs for the subscales were 0.71, 0.71, 0.71, 0.70, and 0.74, respectively. Using CFA to test the construct validity, χ^2^/df = 4.20, TLI = 0.95, CFI = 0.96, RMSEA = 0.07, RMR = 0.01, GFI = 0.89, PNFI = 0.79.

Professional Identity. This study measured professional identity with eighteen items compiled by Wei [48]. The scale has four subscales: occupational values, role values, occupational belonging, and occupational behavior inclination. Sample items of professional identity such as ‘As a teacher, I am a valuable person’ and ‘I am proud of being a teacher’. The responses to statements were rated on a 5-point Likert scale (from 1 = strongly disagree to 5 = strongly agree). Cronbach’s αs for the total scale and subscales were 0.87, 0.86, 0.76, 0.77, and 0.74, respectively. CRs for the total scale and subscales were 0.95, 0.89, 0.82, 0.79, and 0.75, respectively. AVEs for the subscales were 0.52, 0.58, 0.49, 0.50, and 0.51, respectively. Using CFA to test the construct validity, χ^2^/df = 4.31, TLI = 0.92, CFI = 0.94, RMSEA = 0.07, RMR = 0.03, GFI = 0.92, PNFI = 0.72.

Job Satisfaction. This study measured job satisfaction with three items compiled by Liu [49]. Sample items of job satisfaction such as ‘In general, I am satisfied with my work’ and ‘In general, I like working here’. The responses to statements were rated on a 5-point Likert scale (from 1 = strongly disagree to 5 = strongly agree). Cronbach’s α for the scale was 0.92. CR for the scale was 0.96. AVE for the scale was 0.88.

### 4.4. Statistical Analysis

Amos 21.0 software was used for the validity test, and SPSS 24.0 software was used for the reliability analysis, descriptive statistics, correlation analysis, and hypothesis test.

## 5. Results

### 5.1. Descriptive Statistics and Correlation Analysis

The mean value, standard deviation, and Pearson correlation matrix are shown in Table 1. The correlation analysis showed that school atmosphere, psychological capital, and job satisfaction were positively correlated. There were significant positive correlations among school atmosphere, professional identity, and job satisfaction. School atmosphere, psychological capital, professional identity, and job satisfaction were positively correlated with each other. The correlation between variables supports subsequent hypothesis testing. In addition, gender was significantly positively correlated with school atmosphere, professional identity, and job satisfaction, so gender should be used as a control variable in hypothesis testing.

### 5.2. Hypothesis Testing

To reduce statistical errors, gender, school type, teaching subject, and school location were used as the control variables. Before the mediation analysis, we examined the multicollinearity of the model. When the outcome variables were psychological capital, professional identity, and job satisfaction, the VIF values of each antecedent variable were less than the critical value of 10, indicating that there was no serious collinearity in the model, and the results were acceptable.

The total effect of school atmosphere on job satisfaction was tested, and it was found that school atmosphere had a significant positive predictive effect on job satisfaction (*B* = 0.87, *SE* = 0.05, *t* = 18.58, *p* < 0.001). The result supports Hypothesis 1.

Hypotheses 2–4 were tested with Hayes’ PROCESS macro based on 5000 bootstrap samples. The results are shown in Figure 2 and Table 2. The results showed that school atmosphere had a significant positive predictive effect on job satisfaction (*B* = 0.41, *SE* = 0.07, *t* = 6.20, *p* < 0.001). School atmosphere had a significant positive predictive effect on psychological capital (*B* = 0.85, *SE* = 0.03, *t* = 32.57, *p* < 0.001). The positive predictive effect of psychological capital on job satisfaction was not significant (*B* = 0.01, *SE* = 0.06, *t* = 0.15, *p* = 0.88). The mediating effect test showed that 95%CI [−0.12, 0.15] included 0, indicating that psychological capital did not mediate the relationship between school atmosphere and job satisfaction. The ratio of the mediating effect size to the total effect size was 1.15%. The result did not support Hypothesis 2a,b.

School atmosphere had a significant positive predictive effect on professional identity (*B* = 0.29, *SE* = 0.04, *t* = 7.21, *p* < 0.001). Professional identity had a significant positive predictive effect on job satisfaction (*B* = 1.03, *SE* = 0.06, *t* = 16.92, *p* < 0.001), and the mediating effect test showed that 95%CI [0.22, 0.40] did not include 0, indicating that professional identity played a mediating role between school atmosphere and job satisfaction. The ratio of the mediating effect size to the total effect size was 34.48%. The result supports Hypothesis 3a,b.

The positive predictive effect of psychological capital on professional identity was significant (*B* = 0.17, *SE* = 0.04, *t* = 4.54, *p* < 0.001). The mediating effect test showed that 95%CI [0.09, 0.21] did not include 0. The results showed that psychological capital and professional identity played a chain mediating role between school atmosphere and job satisfaction, and the ratio of the mediating effect size to the total effect size was 17.24%. The result supports Hypothesis 4.

## 6. Discussion

### 6.1. Effect of School Atmosphere on Job Satisfaction

This study explored the relationship between school atmosphere and job satisfaction perceived by primary and secondary school teachers. The results showed a significant positive correlation between school atmosphere and job satisfaction, which supports Hypothesis 1. School atmosphere perceived by teachers can make teachers believe that education can bring them opportunities for professional development and enable them to experience respect, happiness, and a sense of achievement. Then, the teachers’ job satisfaction will eventually be improved [50]. Therefore, the school atmosphere plays a critical role in improving the job satisfaction of primary and secondary school teachers. The school’s respect, support, and care for teachers as well as harmonious colleague relations, will, to a certain extent, improve the job satisfaction of primary and secondary school teachers.

### 6.2. The Mediating Role of Psychological Capital and Professional Identity

Although this study did not find the predictive role of psychological capital on job satisfaction, psychological capital can affect job satisfaction through the complete mediation role of professional identity. This is consistent with the research results of [45]. It means that a good school atmosphere can improve individual psychological capital. However, a high level of psychological capital cannot improve the job satisfaction of primary and secondary school teachers. Specifically, when individuals are negatively affected by the external environment, individuals with high psychological capital adapt to the environment through active resource seeking (such as seeking support from colleagues) and compensation behavior (such as avoidance and confrontation) [51]. This shows that individuals with high-level psychological capital are not only passively stimulated, but flexibly adopt corresponding behavioral strategies according to their psychological conditions and abilities to avoid further loss of their psychological resources [52]. In this study, only when teachers with high-level psychological capital have a high-level professional recognition of the teacher’s career will they choose positive resource-seeking behavior in the face of difficulties and obstacles in education and teaching, and feel the professional pride brought by education and teaching activities many times in their education and teaching practice, and finally achieve an improvement in their job satisfaction. However, those teachers with a high level of psychological capital but do not agree with the teaching profession themselves will choose to avoid, confront, and/or other compensation behaviors in the face of possible further loss of their resources such as resignation and negative treatment of work.

Meanwhile, this study found that professional identity partially mediated between school atmosphere and job satisfaction, which supports Hypothesis 3. This is consistent with the previous research results: a good school atmosphere promotes the improvement in the teachers’ professional identity [53], thereby improving the teachers’ job satisfaction [54]. This shows that a fair and just school management system, support from superiors for teachers, encouragement from colleagues, and the adoption of the teachers’ views by the organizations can make it easier for teachers to meet their unique needs and have high-level recognition of their current teaching career [55], thus improving their job satisfaction.

### 6.3. The Chain Mediating Role of Psychological Capital and Professional Identity

This study further found that psychological capital can positively predict professional identity, and psychological capital and professional identity play a chain mediating role between school atmosphere and job satisfaction, which supports Hypothesis 4. This is consistent with previous research results [56] that shows that a positive school atmosphere can strengthen the resilience and confidence of primary and secondary school teachers when facing difficulties and challenges, make their professional identity more positive, and thus improve their job satisfaction. Therefore, by integrating the appropriate school environment, mastering the skills to face difficulties, and maintaining a good attitude will improve the psychological capital and positive professional identity of primary and secondary school teachers and finally help to improve their job satisfaction.

### 6.4. Practical Implications

The results of this study show that a positive school atmosphere has a significant positive predictive effect on the job satisfaction of teachers. The psychological capital of primary and secondary school teachers had no significant predictive effect on their job satisfaction; professional identity played the part of a mediating role in the influence of school atmosphere on the job satisfaction of primary and secondary school teachers.

School is the central place for teachers and students to teach and learn. It is an important environmental factor that affects the teachers’ professional development in primary and secondary schools. It has a relatively lasting and stable impact on the teachers’ behavior. Therefore, in online teaching, school administrators should establish a smooth channel for teachers to participate in school decision-making and encourage teachers to put forward different demands. At the same time, the school should often hold exchange meetings to provide teachers with mutual trust and cooperation work exchange opportunities to meet their professional development needs. Finally, a fair, simple, and transparent teacher evaluation mechanism in the online teaching process and timely feedback from school administrators will also help improve the job satisfaction of teachers.

The research results are of great significance to the development of primary and secondary school teachers. First, professional identity is an essential basic factor for the development of primary and secondary school teachers and the core of the teaching career, reflecting the common needs of humans and society. From the perspective of social culture, professional identity affects the individuals’ basic work attitudes, cognition, and feelings [57]. Therefore, during online teaching, the school can improve the psychological capital level of primary and secondary school teachers by opening additional online technical training, psychological counseling, and mutual aid meeting, mobilizing their enthusiasm for work and enabling them to continue learning and improve in their work. At the same time, through a good management system, the school can develop various online teacher–student interaction activities to promote a friendly relationship between teachers and students and strengthen effective communication and exchange during the epidemic. All this is also very helpful in improving the professional identity of primary and secondary school teachers. The third point is about the distribution of workload and time. The reason may be that teachers are busy repeating complex non-teaching work every day; especially during COVID-19, the workload increased energy and physical consumption, and the enthusiasm for teaching work gradually decreased.

Therefore, according to their characteristics, schools should properly correct the existing teachers’ work content, reduce the non-teaching work content, fully reflect the teachers’ self-worth, improve the teachers’ subjective initiative, promote teacher unity, and create a good school atmosphere.

## 7. Conclusions

This study answered how school atmosphere affects the job satisfaction of primary and secondary school teachers by building a chain intermediary model. The school atmosphere can affect the job satisfaction of primary and secondary school teachers through partly mediating the role of professional identity and the chain mediating role of psychological capital and professional identity. The part mediating role of professional identity revealed the influence mechanism of school atmosphere on the job satisfaction of primary and secondary school teachers from the perspective of individual psychological state and professional emotional experience. In addition, the chain intermediation of psychological capital and professional identity showed that professional identity mediates the relationship between environmental factors and individual job satisfaction and is constrained by the level of individual psychological capital while positively affecting individual job satisfaction.

Therefore, this study proposes that schools adopt more effective school management strategies to build a positive school atmosphere to improve the teachers’ psychological capital and professional identity to solve the practical problem of low job satisfaction among primary and secondary school teachers.

## 8. Limitations

Finally, although the results of this study confirm that there is a relationship between school atmosphere, the teachers’ psychological capital, the teachers’ professional identity, and their job satisfaction, we must note that the research results based on the survey of primary and secondary school teachers in China are not very scalable when the social systems of China and most countries in the world are different. It can only provide some reference for future research on the job satisfaction of teachers.

## Figures and Tables

**Figure 1 behavsci-13-00001-f001:**
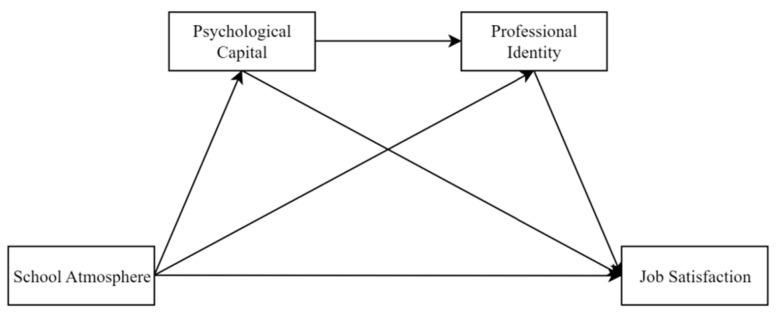
Theoretical model.

**Figure 2 behavsci-13-00001-f002:**
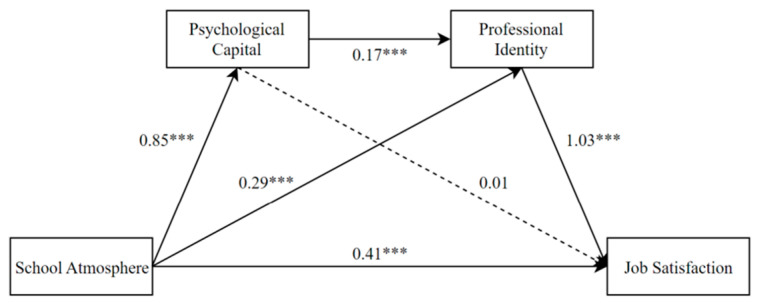
Mediating role of psychological capital and professional identity.

**Table 1 behavsci-13-00001-t001:** Descriptive statistics and correlation analysis of variables.

	M ± SD	1	2	3	4	5	6	7
1. Gender	1.82 ± 0.39							
2. School Type	1.66 ± 0.79	−0.19 **						
3. Teaching Subject	1.42 ± 0.68	−0.15 **	0.28 **					
4. School Location	2.09 ± 0.80	−0.02	−0.13 **	−0.10 **				
5. School Atmosphere	4.51 ± 0.53	0.12 **	−0.09 *	−0.05	<0.01			
6. Psychological Capital	4.42 ± 0.57	<0.01	−0.06	−0.01	−0.04	0.78 **		
7. Professional Identity	4.57 ± 0.41	0.10 *	0.01	0.02	0.03	0.56 **	0.53 **	
8. Job Satisfaction	4.43 ± 0.77	0.10 *	0.01	0.02	−0.01	0.59 **	0.51 **	0.71 **

Note. Gender: male = 1, female = 2; School type: primary school = 1, junior high school = 2, high school = 3; Teaching subject: Chinese, math, or English = 1, politics, geography, history, physics, chemistry, or biology = 2, music, sports, or art = 3; School location: countryside = 1, city = 2; * *p* < 0.05, ** p < 0.01.

**Table 2 behavsci-13-00001-t002:** Mediating role of psychological capital and professional identity.

		Estimate	95%CI	Effect Size
Direct Effect	School Atmosphere → Job Satisfaction	0.41	[0.28, 0.54]	47.13%
Indirect Effect	School Atmosphere → Psychological Capital → Job Satisfaction	0.01	[−0.12, 0.15]	1.15%
	School Atmosphere → Professional Identity → Job Satisfaction	0.30	[0.22, 0.40]	34.48%
	School Atmosphere → Psychological Capital → Professional Identity → Job Satisfaction	0.15	[0.09, 0.21]	17.24%
Total Indirect Effect		0.46	[0.33, 0.60]	52.87%
Total Effect		0.87	[0.78, 0.96]	100.00%

## Data Availability

Data will be provided by the authors on request.

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
