# Peer review of "The Influence of School Atmosphere on Chinese Teachers’ Job Satisfaction: The Chain Mediating Effect of Psychological Capital and Professional Identity"

_behavsci, 2022, doi:10.3390/bs13010001_

Round 1

Reviewer 1 Report

Your work underlined an important issue in education with the title “School atmosphere and job satisfaction: The role of psychological capital and professional identity”. In your study, you created the model and made the analysis accordingly. However, some corrections need to be made and I tried to list them one by one below.

- In the introductory part of your abstract, you should mention the subject and the problem with one or two sentences. It is not very convenient to enter with a direct research sentence.

- It is not clear how, with which method and program you analyzed the study in the summary section. Please write them all in one short sentence.

- Your sentence starting with 648 is not written in an academic language. It needs to be restated.

- Why didn't you mention direct effects? However, they are also very important findings, aren't they?

- A single paragraph written under the title of 1. Introduction is not enough. The study's research question and the theory on which it is based should be written here. A broader introduction should be written. It should consist of at least two paragraphs.

- In the literature section, there should be hypotheses for direct analysis. Because without these effects being understood, there will be no mediation effect.

- The resources used in the literature part are insufficient.

- There are serious English spelling errors in the literature section. It should be corrected.

- In the literature section, the explanations about job satisfaction are not written enough. It should be supported and informed by more resources than current literature.

- Why are the CFA values ​​not written for job satisfaction?

- At least two sample items of the scales should be written.

- How the evaluations of the measures should be written. For example, as the total value of the scale increases, the positive school atmosphere increases.

- What demographic questions did you ask? Unwritten. Is it just gender? Why?

- The discussion section is insufficient. It will be more understandable if you make the subtitles as Direct, indirect, and Chain Mediating or Serial Mediating. At the end of the discussion, you can open a sub-title for the evaluation of the research question you will write in the introduction part.

- Limitations title and content are not written.

- You can add Implications to the Conclusion.  

Author Response

Dear reviewer:

We thank you for these positive comments. We believe that we have been able to address your comments and suggestions and that our paper has been substantially improved as a result. According to the great suggestions, we followed these suggestions and made subsequent changes to the manuscript. We marked the changed contents in red font. Below, we indicate how we responded to each of your comments.

Comment 1:

In the introductory part of your abstract, you should mention the subject and the problem with one or two sentences. It is not very convenient to enter with a direct research sentence.

Response to comment 1:

Thank you for your comment. We have rewritten this part according to your comment: Until today, the impact of organizational climate on job satisfaction has still attracted the attention of researchers in the field of education to help decision-makers and school leaders improve teachers' commitment, professional identity, and job satisfaction through the construction of the school environment.

Comment 2:

 It is not clear how, with which method and program you analyzed the study in the summary section. Please write them all in one short sentence.

Response to comment 2:

Thank you for your comment. We have rewritten this part according to your comment: The school atmosphere, psychological capital, professional identity, and job satisfaction scales were used to investigate 648 primary and secondary school teachers in China. The Bootstrap method was used to test the mediating effect.

Comment 3:

Your sentence starting with 648 is not written in an academic language. It needs to be restated.

Response to comment 3:

Thank you for your comment. We have rewritten this part according to your comment: The school atmosphere, psychological capital, professional identity, and job satisfaction scales were used to investigate 648 primary and secondary school teachers in China.

Comment 4:

Why didn't you mention direct effects? However, they are also very important findings, aren't they?

Response to comment 4:

Thank you for your comment. As you said, they are also very important findings. We have rewritten this part according to your comment: The results of this study found school atmosphere positively predicts teachers’ job satisfaction; Psychological capital does not mediate between school atmosphere and job satisfaction; Professional identity plays a mediating role between school atmosphere and job satisfaction; Psychological capital and professional identity play a chain mediating role between school atmosphere and job satisfaction.

Comment 5:

A single paragraph written under the title of 1. Introduction is not enough. The study's research question and the theory on which it is based should be written here. A broader introduction should be written. It should consist of at least two paragraphs.

Response to comment 5:

Thank you for your comment. Other reviewers also made the same comment. We have added a broader introduction.

After the outbreak of novel coronavirus pneumonia in 2019, Chinese schools suddenly switched from the traditional school learning mode to the online education mode. During online teaching, as teachers need to learn new digital technologies to help online teaching more smoothly, teachers' workload has increased compared with that before, and the quality of teachers' work has become increasingly demanding [1]. When the pressure from the high workload and level of job requirements acts on teachers, teachers are likely to have a sense of frustration due to their limited resources and even directly lead to a significant decline in their job satisfaction. However, when teachers' job satisfaction is seriously reduced, it is more likely to cause teachers to implement teaching activities inefficiently and affect the regular teaching order [2].

The school atmosphere is a common psychological perception. This common psychological perception can promote mutual trust and help managers and teachers, teachers and teachers, to achieve the training goals set by the school for students and improve teachers' job satisfaction and sense of teaching mission. However, in the form of online teaching, the promotion of school atmosphere for teachers is very limited.

As far as practical work is concerned, due to online teaching, teachers must constantly fight against technical failures during video conferences, which makes them feel that their work tasks are getting heavier and heavier. Their work tension is be-coming more and more apparent. However, teachers cannot properly solve these problems [3]. Because of the epidemic, they cannot promptly seek support and comfort from schools and colleagues as they did before the epidemic. Teachers cannot ease tension for such a long time, likely leading to the continuous loss of psychological capital. Over time, teachers are prone to insomnia, tension, irritability, fatigue, and other negative emotions, leading to a decline in their professional identity and, ultimately, their job satisfaction. In the long run, this will affect the regular operation of school teaching, especially in the current unique situation. Therefore, it is necessary to explore the role of psychological capital and professional identity in the impact of school atmosphere on teachers' job satisfaction.

In the research on primary and secondary school teachers' professional problems, only some studies have simultaneously examined the impact of environmental factors and personal factors on teachers' job satisfaction. This study further explored the mediating role of psychological capital and professional identity of primary and second-ary school teachers in the school atmosphere affecting teachers' job satisfaction.

To sum up, a good school atmosphere can strengthen the trust between managers and teachers and reach a consensus so that teachers can persevere when encountering obstacles and come up with solutions at will, thus improving their psychological capital level [4,5]. Furthermore, teachers with high-level psychological capital can invest more energy and time in their work to learn and develop their potential, experience the fun of teaching, effectively improve their professional identity [6], and ultimately make them more satisfied with their work [7,8]. Therefore, it is not enough to just dis-cuss job satisfaction to improve the current job satisfaction of primary and secondary school teachers. It also needs to consider the impact of the school atmosphere, teachers' psychological capital, and professional identity. In order to verify the feasibility of this measure, it is necessary to study the job satisfaction of primary and secondary school teachers and determine the impact of school atmosphere, psychological capital, and professional identity on job satisfaction. On this basis, it is possible to propose practical measures to ensure the effectiveness of teachers' work and provide a basis for further improving and managing the job satisfaction of primary and secondary school teachers.

Comment 6:

 In the literature section, there should be hypotheses for direct analysis. Because without these effects being understood, there will be no mediation effect.

Response to comment 6:

Thank you for your comment. The hypotheses for direct analysis are very important. We have specially added a literature review section to provide a theoretical basis for the establishment of assumptions. In addition, we use many theoretical foundations to demonstrate the hypothesis of direct path.

The school atmosphere is the teacher’s subjective experience of the school’s norms, objectives, values, interpersonal relationships, teaching and learning practices, and organizational structure. It is an environmental feature that has a relatively lasting and stable impact on teachers’ behavior [19]. School atmosphere can be divided into organization management, team cooperation, teaching efficiency, and resource use [20]. Research shows that individuals’ perceptions of the organizational environment significantly affect their job satisfaction [21,22], and further affect their job engagement and performance [23]. Social information processing theory believes that the social environment in which individuals live can affect their attitudes and beliefs and thus affect their perception of working conditions and states [24]. In addition to individual personality, the job satisfaction of organizational members is also affected by their perceived organizational culture. A positive organizational culture helps improve organizational members’ job satisfaction [25]. Therefore, as teachers perceive organizational culture in their work situations, the school atmosphere can affect teachers’ subjective feelings and job satisfaction. Therefore, this study hypothesizes that:

Hypothesis 1: School atmosphere positively predicts teachers’ job satisfaction.

Comment 7:

The resources used in the literature part are insufficient.

Response to comment 7:

Thank you for your comment. It is very important. In view of this problem, we have made a lot of modifications to the literature section to make it more sufficient. On the basis of the introduction section of the original manuscript, we have added a new ‘Literature Review and Theoretical Background’ section, specifically using a large number of literature to write the research content of this study. In order to make the literature part more sufficient, our word count has also increased by about 2000 words.

Comment 8:

There are serious English spelling errors in the literature section. It should be corrected.

Response to comment 8:

Thank you for your comment. We have corrected all known spelling errors. If there are other spelling errors, we look forward to your further suggestions.

Comment 9:

 In the literature section, the explanations about job satisfaction are not written enough. It should be supported and informed by more resources than current literature.

Response to comment 9:

Thank you for your comment. We have rewritten this part according to your comment: Teachers’ job satisfaction has many important and far-reaching implications [9]. It refers to the overall feelings of individual teachers regarding their work conditions [9]. It can significantly predict teacher retention and performance [10]. The job satisfaction of primary and secondary school teachers is a crucial factor affecting their work attitude and behavior [11]. At present, scholars around the world have accumulated a large number of fruitful results in research on the job satisfaction of primary and secondary school teachers. This study focuses on the environmental factors and personal characteristics that affect the job satisfaction of primary and secondary school teachers [12]. For example, Zeeshan believes that the main factors that affect teacher satisfaction are school management, work environment, salary, colleague relations, work itself, safety, and other factors [13]. Lam investigated the factors that affected teachers' job satisfaction and proposed that school size, working hours, leader-ship, work achievements, and other factors will affect teachers' job satisfaction [14]. Shaukat discussed the impact of gender, age, background, teaching experience, and professional qualifications of special education teachers in Pakistan on their job satisfaction. The results show that teachers' gender, age, educational background, teaching experience, and other characteristics significantly impact job satisfaction [15]. Teachers’ job satisfaction affects their work status, teaching effectiveness, and quality of talent training [16].

Currently, the job satisfaction of primary and secondary school teachers in China is generally low [17], which seriously hinders the enthusiasm and enthusiasm of teachers and the smooth development of teaching activities and restricts the practical improvement of education quality [18]. Therefore, it is crucial for primary and secondary school teachers to improve job satisfaction, whether from the perspective of strengthening the construction of teachers, improving the working status of teachers, or the growth and development of students.

Comment 10:

Why are the CFA values not written for job satisfaction?

Response to comment 10:

Thank you for your comment. The reason why we did not calculate the CFA of job satisfaction is that when the variable is a single factor model, its Cronbach’s α reflects the construct validity of CFA. In addition, there are only 3 items of job satisfaction, and the 3 items cannot be subject to CFA. Because its df = 0, the items for CFA must be greater than or equal to 4. As a single factor structure, the Cronbach’s α of job satisfaction itself reflects the structural validity. Cronbach’s α for the scale was 0.92. Therefore, its construct validity is also good.

Comment 11:

 At least two sample items of the scales should be written.

Response to comment 11:

Thank you for your comment. We have added relevant sample items. Sample items of school atmosphere, such as ‘The school will praise and encourage active and excellent teachers’ and ‘The school has many initiatives to encourage teaching reform and innovation’. Sample items of psychological capital, such as ‘I am optimistic about what will happen to my work in the future’ and ‘I usually take the pressure at work calmly’. Sample items of professional identity, such as ‘As a teacher, I am a valuable person’ and ‘I am proud of being a teacher’. Sample items of job satisfaction, such as ‘In general, I am satisfied with my work’ and ‘In general, I like working here’.

Comment 12:

 How the evaluations of the measures should be written. For example, as the total value of the scale increases, the positive school atmosphere increases.

Response to comment 12:

Thank you for your comment. All scales were scored with Likert 5 points (from 1= strongly disagree to 5 = strongly agree). Therefore, as the total value of the scale increases, the positive school atmosphere increases. We have supplemented the evaluations of the measures in the manuscript.

Comment 13:

What demographic questions did you ask? Unwritten. Is it just gender? Why?

Response to comment 13:

In addition to the demographic variable of gender, we also measured the school type (such as primary school, junior high school, high school), subjects, and school location (rural or urban). We calculated the correlation coefficients between these demographic variables and school atmosphere, psychological capital, professional identity and job satisfaction. The results show that only the school type has a significant correlation with the school atmosphere, and other correlation coefficients are not significant, indicating that these demographic variables have a small impact on the structural equation model and can not be used as control variables. So we didn't write it in the manuscript. Moreover, gender is significantly positively correlated with school climate, professional identity, and job satisfaction, so gender only be used as a control variable in hypothesis testing.

Comment 14:

 The discussion section is insufficient. It will be more understandable if you make the subtitles as Direct, indirect, and Chain Mediating or Serial Mediating. At the end of the discussion, you can open a sub-title for the evaluation of the research question you will write in the introduction part.

Response to comment 14:

Thank you for your comment. To explain the practical value of the research mentioned in the introduction, we have added the Practical Implication part and expanded the Discussion part according to your comment.

It is worth mentioning that, considering that the mediation effect of psychological capital is not significant in this model, we still decide to put all mediation effects in this model under the same title. But we segment them to help readers better understand all mediations.

The first paragraph is used to explain the complete mediation effect of psychological capital in the model.

The second paragraph is used to explain the mediation effect of professional identity in the model.

The third paragraph is used to explain the chain mediation effect in the model.

Practical Implication:The results of this study show that a positive school atmosphere has a significant positive predictive effect on teachers' job satisfaction. The psychological capital of primary and secondary school teachers has no significant predictive effect on their job satisfaction; Professional identity plays a part of mediating role in the influence of school atmosphere on the job satisfaction of primary and secondary school teachers.

School is the central place for teachers and students to teach and learn. It is an important environmental factor that affects teachers' professional development in primary and secondary schools. It has a relatively lasting and stable impact on teachers' behavior. Therefore, in online teaching, school administrators should establish a smooth channel for teachers to participate in school decision-making and encourage teachers to put forward different demands. At the same time, the school should often hold exchange meetings to provide teachers with mutual trust and cooperation work exchange opportunities to meet their professional development needs. Finally, a fair, simple, and transparent teacher evaluation mechanism in the online teaching process and timely feedback from school administrators will also help improve teachers' job satisfaction.

The research results are of great significance to the development of primary and secondary school teachers. First, professional identity is an essential basic factor for the development of primary and secondary school teachers and the core of the teaching career, reflecting the common needs of humans and society. From the perspective of social culture, professional identity affects individuals' basic work attitudes, cognition, and feelings [58]. Therefore, during online teaching, the school can improve the psychological capital level of primary and secondary school teachers by opening additional online technical training, psychological counseling, and mutual aid meeting, mobilizing their enthusiasm for work and enabling them to continue learning and im-prove in their work. At the same time, through a good management system, the school can develop various online teacher-student interaction activities to promote a friendly relationship between teachers and students and strengthen effective communication and exchange during the epidemic. All this is also very helpful in improving the professional identity of primary and secondary school teachers. The third point is about the distribution of workload and time. The reason may be that teachers are busy repeating complex non-teaching work every day, especially during COVID-19; the workload increases energy and physical consumption, and the enthusiasm for teaching work gradually decreases.

Therefore, according to their characteristics, schools should properly correct the existing teachers' work content, reduce teachers' non-teaching work content, fully reflect teachers' self-worth, improve teachers' subjective initiative, promote teachers' unity, and create a good school atmosphere.

Comment 15:

Limitations title and content are not written. You can add Implications to the Conclusion.  

Response to comment 15:

Thank you for your comment. We have added the Limitations part according to your comment:

Finally, although the results of this study confirm that there is a relationship be-tween school atmosphere, teachers' psychological capital, teachers' professional identity, and their job satisfaction, we must note that the research results based on the sur-vey of primary and secondary school teachers in China are not very scalable when the social systems of China and most countries in the world are different. It can only pro-vide some reference for future research on teachers' job satisfaction.

Reviewer 2 Report

The title of the paper should be reformulated. For example "School atmosphere influence on psychological capital and professional identity improving job satisfaction"  as to emphaisze the link between variables...

The aim is clear: Evaluating the influence of school atmosphere on teachers’ job satisfaction and the testing mediation role of psychological capital and professional identity on job satisfaction 

The abstract should be rearranged to contain context, methods, results, and conclusions in an academically fluid manner, without containing the words "background, methods, results, and conclusion

The introduction should present the state of the art and the research questions. The authors designed the introduction as would be a literature review background. These are 2 different sections. The theoretical model should be presented at the end of the second section "Theoretical background" followed by the research hypothesis.

The authors meet the requirements of the methodology overall, but they might emphasize that this study is representative for Taiyuan. It might be added in the title too. 

The study methods are valid and reliable and can be replicated, but the authors should present the survey in an Annex. as to be able to completely understand the study

Is very important that the authors to received IRB acceptance and the agreement of the respondents. “Ethical standards were applied in the conduct of the study  ”.

In the results section, tables and figures are relevant and clearly presented. Titles, columns, and rows labeled are correctly and clearly presented.  I would advise the authors to add a multicollinearity analysis and a discriminant validity having in mind that all Cronbach Alpha, Composite reliability, AVE, are extremely high.

The authors tried to interpret the data. The text in the results is not repetitive.

The results are discussed from multiple angles and placed into context without being overinterpreted.

Mainly the conclusions supported by references or results.

Please add new opinions that appeared in the articles published in the last 3 years because they also discuss the role and impact of technology as the pandemic forced us to teach online. Cite many new articles.

The study design was appropriate to answer the aim but has to be improved with the elements above.

Author Response

Dear reviewer:

We thank you for these important comments. We apologize that the prior version of this paper has some errors and shortcomings. It is very helpful to improve the quality of our manuscript. According to the great suggestions, we followed these suggestions and made subsequent changes to the manuscript. We marked the changed contents in red font.

Comment 1:

The title of the paper should be reformulated. For example "School atmosphere influence on psychological capital and professional identity improving job satisfaction" as to emphaisze the link between variables...

Response to comment 1:

Thank you for your comment. Our original title is really vague, which does not reflect the role of psychological capital and professional identity. So, we have reformulated the title. The new title is ‘The influence of school atmosphere on Chinese Teachers’ job satisfaction: The chain mediating effect of psychological capital and professional identity’.

Comment 2:

The abstract should be rearranged to contain context, methods, results, and conclusions in an academically fluid manner, without containing the words "background, methods, results, and conclusion".

Response to comment 2:

Thank you for your comment. We rewrote our abstract.

Until today, the impact of organizational climate on job satisfaction has still attracted the attention of researchers in the field of education to help decision-makers and school leaders improve teachers' commitment, professional identity, and job satisfaction through the construction of the school environment. This study explored the impact of school atmosphere on teachers' job satisfaction and examined the chain mediating role of psychological capital and professional identity. The school atmosphere, psychological capital, professional identity, and job satisfaction scales were used to investigate 648 primary and secondary school teachers in China. The Bootstrap method was used to test the mediating effect. The results showed that school atmosphere had a positive predictive effect on teachers' job satisfaction; Psychological capital does not play a mediating role between school atmosphere and job satisfaction; Professional identity plays a mediating role between school atmosphere and job satisfaction; Psychological capital and professional identity play a chain mediating role between school atmosphere and job satisfaction. Therefore, this study proposes that schools adopt more effective school management strategies to build a positive school atmosphere to improve teachers' psychological capital and professional identity to solve the practical problem of low job satisfaction among primary and secondary school teachers.

Comment 3:

The introduction should present the state of the art and the research questions. The authors designed the introduction as would be a literature review background. These are 2 different sections. The theoretical model should be presented at the end of the second section "Theoretical background" followed by the research hypothesis.

Response to comment 3:

Thank you for your comment, it is important. We have rewritten the introduction to highlight the research background and questions to be solved. And we divided the original introduction into three parts: Literature Review, Theoretical Background and The Present Study. It is worth mentioning that we emphasized the background in the introduction. Specific contents are as follows:

After the outbreak of novel coronavirus pneumonia in 2019, Chinese schools suddenly switched from the traditional school learning mode to the online education mode. During online teaching, as teachers need to learn new digital technologies to help online teaching more smoothly, teachers' workload has increased compared with that before, and the quality of teachers' work has become increasingly demanding [1]. When the pressure from the high workload and level of job requirements acts on teachers, teachers are likely to have a sense of frustration due to their limited resources and even directly lead to a significant decline in their job satisfaction. However, when teachers' job satisfaction is seriously reduced, it is more likely to cause teachers to implement teaching activities inefficiently and affect the regular teaching order [2].

In the research on primary and secondary school teachers' professional problems, only some studies have simultaneously examined the impact of environmental factors and personal factors on teachers' job satisfaction. This study further explored the mediating role of psychological capital and professional identity of primary and secondary school teachers in the school atmosphere affecting teachers' job satisfaction.

Comment 4:

The authors meet the requirements of the methodology overall, but they might emphasize that this study is representative for Taiyuan. It might be added in the title too. 

Response to comment 4:

Thank you for your comment. There are great differences among teachers in different cultures. However, it is too subtle to focus the subjects on one city and many people may not know where Taiyuan is. Since our subjects are Chinese teachers, we so we wrote about Chinese teachers in the topic

Comment 5:

The study methods are valid and reliable and can be replicated, but the authors should present the survey in an Annex. as to be able to completely understand the study

Response to comment 5:

Thank you for your comment. We are ready to upload relevant contents to the system as Annex.

Comment 6:

Is very important that the authors to received IRB acceptance and the agreement of the respondents. “Ethical standards were applied in the conduct of the study”.

Response to comment 6:

Thank you for your comment. We have added the Ethical Consideration part according to your comment:

This study was conducted following the Declaration of Helsinki (2002) and Measures for Ethical Review of Biomedical Research Involving Humans, Ministry of Health, China. The protocol was approved by the Ethics Committee of Shanxi Normal University

Comment 7:

I would advise the authors to add a multicollinearity analysis and a discriminant validity having in mind that all Cronbach Alpha, Composite reliability, AVE, are extremely high.

Response to comment 7:

Thank you for your comment. Firstly, the collinearity problem in structural equation models often shows that the standardized factor load or path coefficient is greater than 1. However, all our standardized factor load and path coefficients are less than 1, indicating that there is no serious collinearity problem.

Secondly, we also calculated Cronbach’s α, CR, and AVE. The results are as follows:

School Atmosphere. Cronbach’s αs for the total scale and subscales were 0.94, 0.81, 0.87, 0.91, and 0.87, respectively. CRs for the total scale and subscales were 0.95, 0.82, 0.87, 0.92, and 0.89, respectively. AVEs for the subscales were 0.52, 0.53, 0.62, 0.79, and 0.74, respectively.

Psychological Capital. Cronbach’s αs for the total scale and subscales were 0.98, 0.94, 0.93, 0.94, and 0.94, respectively. CRs for the total scale and subscales were 0.98, 0.94, 0.94, 0.93, and 0.95, respectively. AVEs for the subscales were 0.71, 0.71, 0.71, 0.70, and 0.74, respectively.

Professional Identity. Cronbach’s αs for the total scale and subscales were 0.87, 0.86, 0.76, 0.77, and 0.74, respectively. CRs for the total scale and subscales were 0.95, 0.89, 0.82, 0.79, and 0.75, respectively. AVEs for the subscales were 0.52, 0.58, 0.49, 0.50, and 0.51, respectively.

Job Satisfaction. Cronbach’s α for the scale was 0.92. CR for the scale was 0.96. AVE for the scale was 0.88.

Comment 8:

Please add new opinions that appeared in the articles published in the last 3 years because they also discuss the role and impact of technology as the pandemic forced us to teach online. Cite many new articles.

Response to comment 8:

Thank you for your comment. Your comment is very important to us. We added new opinions that appeared in the articles published in the last three years and combined the COVID-19 epidemic with the research background. In the total number of cited literature, the published literature in the recent three years accounts for almost half of the total number of cited literature.

Reviewer 3 Report

1. The paper title should specify the scope of the research subject.

2. Abstract should describe more specifically the characteristics of the subject, such as country or province.

3. In the background, meaning and importance of the study are not understood in the Introduction.

4. The research model lacks theoretical support.

5. The sampling process should be described in more detail.

6. In Results, item analysis, reliability and validity analysis, and model fit analysis are missing.

7. Conclusion should be supplemented with contributions, recommendations, study limitations and suggestions for future research.

8. Overall, this paper is not complete. Many paragraphs do not explain why this is necessary.

Author Response

Dear reviewer:

We thank you for these important comments. We apologize that the prior version of this paper has some errors and shortcomings. It is very helpful to improve the quality of our manuscript. According to the great suggestions, we followed these suggestions and made subsequent changes to the manuscript. We marked the changed contents in red font.

Comment 1:

 The paper title should specify the scope of the research subject.

Response to comment 1:

Thank you for your comment. It is very important. There are great differences among teachers in different cultures. Since our subjects are Chinese teachers, we have reflected Chinese teachers in the paper title.

Comment 2:

Abstract should describe more specifically the characteristics of the subject, such as country or province.

Response to comment 2:

Thank you for your comment. We rewrote our abstract according to your comment. Our research object is primary and secondary school teachers from China. We have pointed out in the new Abstract.

Comment 3:

In the background, meaning and importance of the study are not understood in the Introduction.

Response to comment 3:

Thank you for your comment. We rewrote our Introduction according to your comment:

After the outbreak of novel coronavirus pneumonia in 2019, Chinese schools suddenly switched from the traditional school learning mode to the online education mode. During online teaching, as teachers need to learn new digital technologies to help online teaching more smoothly, teachers' workload has increased compared with that before, and the quality of teachers' work has become increasingly demanding [1]. When the pressure from the high workload and level of job requirements acts on teachers, teachers are likely to have a sense of frustration due to their limited resources and even directly lead to a significant decline in their job satisfaction. However, when teachers' job satisfaction is seriously reduced, it is more likely to cause teachers to implement teaching activities inefficiently and affect the regular teaching order [2].

The school atmosphere is a common psychological perception. This common psychological perception can promote mutual trust and help managers and teachers, teachers and teachers, to achieve the training goals set by the school for students and improve teachers' job satisfaction and sense of teaching mission. However, in the form of online teaching, the promotion of school atmosphere for teachers is very limited.

As far as practical work is concerned, due to online teaching, teachers must constantly fight against technical failures during video conferences, which makes them feel that their work tasks are getting heavier and heavier. Their work tension is be-coming more and more apparent. However, teachers cannot properly solve these problems [3]. Because of the epidemic, they cannot promptly seek support and comfort from schools and colleagues as they did before the epidemic. Teachers cannot ease tension for such a long time, likely leading to the continuous loss of psychological capital. Over time, teachers are prone to insomnia, tension, irritability, fatigue, and other negative emotions, leading to a decline in their professional identity and, ultimately, their job satisfaction. In the long run, this will affect the regular operation of school teaching, especially in the current unique situation. Therefore, it is necessary to explore the role of psychological capital and professional identity in the impact of school atmosphere on teachers' job satisfaction.

In the research on primary and secondary school teachers' professional problems, only some studies have simultaneously examined the impact of environmental factors and personal factors on teachers' job satisfaction. This study further explored the mediating role of psychological capital and professional identity of primary and secondary school teachers in the school atmosphere affecting teachers' job satisfaction.

To sum up, a good school atmosphere can strengthen the trust between managers and teachers and reach a consensus so that teachers can persevere when encountering obstacles and come up with solutions at will, thus improving their psychological capital level [4,5]. Furthermore, teachers with high-level psychological capital can invest more energy and time in their work to learn and develop their potential, experience the fun of teaching, effectively improve their professional identity [6], and ultimately make them more satisfied with their work [7,8]. Therefore, it is not enough to just dis-cuss job satisfaction to improve the current job satisfaction of primary and secondary school teachers. It also needs to consider the impact of the school atmosphere, teachers' psychological capital, and professional identity. In order to verify the feasibility of this measure, it is necessary to study the job satisfaction of primary and secondary school teachers and determine the impact of school atmosphere, psychological capital, and professional identity on job satisfaction. On this basis, it is possible to propose practical measures to ensure the effectiveness of teachers' work and provide a basis for further improving and managing the job satisfaction of primary and secondary school teachers.

Comment 4:

The research model lacks theoretical support.

Response to comment 4:

Thank you for your comment. The theoretical support is very important. So, on the basis of the introduction section of the original manuscript, we have added a new ‘Literature Review and Theoretical Background’ section, specifically using a large number of literature to write the theoretical support of this study.

Comment 5:

 The sampling process should be described in more detail.

Response to comment 5:

Thank you for your comment. The sampling process in our original manuscript is really simple, so we describe it in more detail:

A total of 785 questionnaires were distributed to primary and secondary school teachers in China by convenient sampling method. This study set some anti-counterfeiting items, such as “Please directly select ‘strongly agree’”. If the item was selected incorrectly, it would be regarded as invalid data. Moreover, questionnaires with regular answers and too many missing values were also regarded as invalid data. Among 648 distributed questionnaires, 648 valid questionnaires were recovered, with an effective rate of 82.55%.

Comment 6:

In Results, item analysis, reliability and validity analysis, and model fit analysis are missing.

Response to comment 6:

Thank you for your comment. We have made the above analysis according to the comment. Firstly, we tested the factor loads of all the questions, and the results showed that the standardized factor loads of all the questions were greater than 0.3. Secondly, we calculated the Cronbach’s α and CR, and the results showed that all Cronbach’s αs and CRs are greater than 0.7. Thirdly, we did CFA and analyzed AVE. The results showed that all AVEs are greater than 0.5, and indicators of CFA are good (for example, CFA and TLI are greater than 0.9, etc.). It shows that the construct validity of each variable is good and the model fit is good. We wrote the results in the manuscript

Comment 7:

Conclusion should be supplemented with contributions, recommendations, study limitations and suggestions for future research.

Response to comment 7:

Thank you for your comment. We rewrote our Conclusion, Limitations, Practical Implication according to your comment. These parts have changed a lot. I hope you can check them in the article. Thank you for your support

Comment 8:

Overall, this paper is not complete. Many paragraphs do not explain why this is necessary. 

Response to comment 8:

Thank you for your comment. We rewrote our paper, according to your comment.

As you said, our article needs to be completed, which is really a fatal problem in our article. We are very grateful for your valuable opinions on our articles in your busy schedule, which is crucial to our academic growth. At the same time, we hope you can maintain high-quality requirements for us in the future audit. Thank you again for your valuable comments.

In the abstract, we briefly introduced the research background, research purpose, research methods, research conclusions and suggestions.

In the introduction, we introduced the research background and the importance of the research in detail.

In the Literature Review and Theoretical Background, we mainly reviewed the research on school atmosphere, psychological capital, teachers' professional identity and job satisfaction, and put forward research hypotheses based on the analysis of the above contents.

In the Present Study, we describe the current situation and limitations of the research on teachers' job satisfaction, as well as the innovation and significance of this research.

In the Materials and Methods, we introduce the subjects, the specific methods of the study, and the reliability and validity of the research tools in detail.

In the Results, we described the data analysis process, hypothesis testing process and results in detail.

In the Discussion, we explained the direct effect, partial mediation effect, and chain mediation effect in the model in detail and put forward suggestions that can be implemented in practice.

In the Limitations, we pointed out some limitations and shortcomings of the research.

In the Conclusion, we summarized and explained the research conclusions.

Round 2

Reviewer 1 Report

I think there are weaknesses related to the following topics. It would be good to examine them carefully.

- Direct effect hypotheses should also be with mediating variables. Why didn't you? Otherwise, there is no mediation relationship.

- That is, there must be hypotheses for the direct relationships between the independent and the mediators, and between the mediators and the dependent variable.

- The results of the analysis of the control variables do not need to be significant. They can potentially affect analysis results if they are in the model. So it would be nice to use continuous variables as covariates in the analysis.

- You wrote that you corrected the discussion part, but it doesn't seem that way. For example, "The second paragraph is used to explain the mediation effect of professional identity in the model." But the second paragraph is "6.2. The Chain Mediating Role of Psychological Capital and Professional Identity". I think there is a mistake in this part. I think It will be more understandable if you make the subtitles as Direct, indirect, and Chain Mediating or Serial Mediating. At the end of the discussion, you can open a sub-title for the evaluation of the research question you wrote in the introduction part. So it will be better if there are 4 subtitles.

Author Response

Dear reviewer:

We thank Reviewer 1 for these positive comments. We believe that we have been able to address your comments and suggestions and that our paper has been substantially improved as a result. According to the great suggestions, we followed these suggestions and made subsequent changes to the manuscript by using the ‘Track Changes’. Below, we indicate how we responded to each of your comments.

Comment 1: Direct effect hypotheses should also be with mediating variables. Why didn't you? Otherwise, there is no mediation relationship. That is, there must be hypotheses for the direct relationships between the independent and the mediators, and between the mediators and the dependent variable.

Response to comment 1: Thank you for comment, we agree with you very much. Therefore, according to your comment, we have made hypotheses about independent variables and mediators, as well as assumptions about mediators and dependent variables. The hypotheses are as follows

Hypothesis 2a: School atmosphere has a positive predictive effect on teachers' psychological capitalï¼›

Hypothesis 2b: Psychological capital has a positive predictive effect on teachers' job satisfaction.

Hypothesis 3a: School atmosphere has a positive predictive effect on teachers' professional identityï¼›

Hypothesis 3b: Professional identity has a positive predictive effect on teachers' job satisfaction.

We have supplemented these hypotheses in the manuscript.

Comment 2: The results of the analysis of the control variables do not need to be significant. They can potentially affect analysis results if they are in the model. So it would be nice to use continuous variables as covariates in the analysis.

Response to comment 2: Thank you for comment, we agree with you very much. Therefore, we also take school type, teaching subject, and school location as control variables. We re analyzed the data and wrote the latest results in the manuscript.

Comment 3: You wrote that you corrected the discussion part, but it doesn't seem that way. For example, "The second paragraph is used to explain the mediation effect of professional identity in the model." But the second paragraph is "6.2. The Chain Mediating Role of Psychological Capital and Professional Identity". I think there is a mistake in this part. I think It will be more understandable if you make the subtitles as Direct, indirect, and Chain Mediating or Serial Mediating. At the end of the discussion, you can open a sub-title for the evaluation of the research question you wrote in the introduction part. So it will be better if there are 4 subtitles.

Response to comment 3: Thank you for comment, we agree with you very much. According to your suggestion, we will change the subtitle to Direct, indirect, and Chain Mediating. In addition, we added a practical implication section in the discussion. This sub-title specifically discusses the practical enlightenment involved in solving the research problems in the introduction. Therefore, there are four subtitles in the discussion section, respectively on direct effect, indirect effect, chain mediating effect, and practical implication related to research problems.

Reviewer 2 Report

Thank you for your improvements.

I insist to tell you that multicollinearity can appear even when the path coefficient has a value of less than 1.  Sometimes we use different questions that in the subjects' minds have the same meaning, although the researcher can make the difference between them.  You should check the VIF.

Author Response

Dear reviewer:

We thank Reviewer 2 for these important comments. We apologize that the prior version of this paper has some errors and shortcomings. It is very helpful to improve the quality of our manuscript. According to the great suggestions, we followed these suggestions and made subsequent changes to the manuscript by using the ‘Track Changes’.

Comment 1: I insist to tell you that multicollinearity can appear even when the path coefficient has a value of less than 1.  Sometimes we use different questions that in the subjects' minds have the same meaning, although the researcher can make the difference between them.  You should check the VIF.

Response to comment 1: Thank you for comment, we agree with you very much. Therefore, we tested multicollinearity. When the outcome variables were psychological capital, professional identity, and job satisfaction, the VIF values of each antecedent variable were less than the critical value of 10, indicating that there is no serious collinearity in the model, and the results were acceptable. We supplemented the result in the results section.

Reviewer 3 Report

After the revision of this article, the overall quality has improved a lot. It has reached the standard of publishability. Therefore, I will provide my comments to the editor-in-chief.

Author Response

Dear reviewer:

Thank you for giving us the opportunity to publish, thank you very much.

Round 3

Reviewer 1 Report

You made the corrections I wanted. Thank you.